# Optimizing Agronomy Improves Super Hybrid Rice Yield and Nitrogen Use Efficiency through Enhanced Post-Heading Carbon and Nitrogen Metabolism

**Jun Deng** [1], **Jiayu Ye** [1], **Ke Liu** [1,2], **Matthew Tom Harrison** [2], **Xuefen Zhong** [3], **Chunhu Wang** [1], **Xiaohai Tian** [1], **Liying Huang** [1,*] **and Yunbo Zhang** [1]

1   MARA Key Laboratory of Sustainable Crop Production in the Middle Reaches of the Yangtze River/College of Agriculture, Yangtze University, Jingzhou 434025, China
2   Tasmanian Institute of Agriculture, University of Tasmania, Newnham Drive, Launceston, TAS 7248, Australia
3   Agricultural and Rural Bureau of Duodao District, Jingmen 448000, China
*   Correspondence: lyhuang8901@126.com; Tel.: +86-716-806-6541

**Abstract:** The super hybrid rice breeding program in China has raised genetic yield ceilings through morphological improvements and inter-subspecific heterosis. Despite this, little information on the physiological basis underlying this yield transformation exists, and less so on the genotype x environment x management conditions enabling consistent yield gains. Here, we assess grain yield, photosynthetic physiology, and leaf carbon and nitrogen (N) metabolic properties of super rice (Y-liangyou900) under four management practices (i.e., zero-fertilizer control, CK; farmers' practice, FP; high-yield and high-efficiency management, OPT1; and super-high-yield management, OPT2) using a field experiment conducted over five years. Grain yield and agronomic N use efficiency ($AE_N$) of OPT2 were 15% and 10% higher than OPT1, and 30% and 78% higher than FP, respectively. The superior yields of OPT2 were attributed to higher source production capacity, that is, higher leaf photosynthetic rate, carbon metabolic enzyme activity (i.e., AGP and SPS), nitrogen metabolic enzyme activity (i.e., NR, GS, and GOGAT), soluble protein and sugar content, and delayed leaf senescence (the latter due to elevated activity of protective enzyme systems) during grain filling. The higher $AE_N$ of OPT2 was associated with higher activity of leaf carbon metabolic enzyme (i.e., AGP and SPS), nitrogen metabolic enzyme (i.e., NR, GS, GDH, and GOGAT) and protective enzyme (POD) after heading, and lower C/N ratio in grains. We conclude that optimized management (optimized water and fertilizer management with appropriate dense planting) improved grain yield and N use efficiency simultaneously by enhancing post-heading leaf carbon and N metabolism and delayed leaf senescence.

**Keywords:** super hybrid rice; crop management; grain yield; N use efficiency; carbon metabolism; nitrogen metabolism

## 1. Introduction

Rice is one of the most important crops in the world, occupying a strategically important position in the economy and social development of China [1]. Indeed, "basic grain self-sufficiency and absolute food security" has long been a central pillar of China's food security [2]. With rapid population growth and urbanization, however, the per capita area land used for rice cultivation has been gradually decreasing. This suggests that increasing rice yields (production per unit area) may be one of the only sustainable avenues toward ensuring Chinese national food security.

Nitrogen (N) fertilizer, water, and plant density are three important cultivation factors affecting the growth and grain yield of rice [3]. To achieve higher yields, farmers used to grow rice under sufficient conditions of fertilizer and water [4]. At present, the average N input for rice production is 180 kg ha$^{-1}$ in China, which is 75% higher than the world

average [3]. A previous study showed that only 20–30% of nitrogen was absorbed by rice plants under high N fertilizer input, and more than 70% of nitrogen was lost through ammonia volatilization, surface runoff, and nitrate leaching [5]. A large amount of N loss not only reduces the N use efficiency (NUE) of rice and increases economic input, but also leads to a series of environmental problems, such as eutrophication and soil acidification [6]. A reasonable planting density can optimize rice population structure, canopy ventilation and light transmission, promote tiller number and plant growth, and then improve grain yield [7]. In any given year, the management conditions required to obtain maximum yields may differ due to genotype x environment x management interactions [8,9], suggesting that experiments to determine optimal management conditions should be conducted over the long term and across multiple sites [10]. In recent years, super hybrid rice has often been planted with low density due to its strong heterosis in rice production [11]. Although low planting density is beneficial to the growth of individual plants, it may lead to insufficient panicle number in some cases, resulting in low grain yield [12,13]. Higher plant density with reduced N fertilization contributed to early canopy closure, and ultimately a higher biomass production, resource use efficiency, and grain yield in hybrid rice [7,14]. Therefore, the rational regulation of N fertilization, planting density, and water management to improve rice yield and resource use efficiency simultaneously is of great significance in achieving even sustainable rice production in China and beyond.

In order to improve both grain yield and resource use efficiency, previous studies have investigated integrated management techniques of high-yielding and efficient cultivation [15,16]. Integrated management techniques are mainly pairwise or three-way assessments of N fertilizer, planting density, and water management [17]. For example, increasing planting density with reduced N fertilizer input has been reported to increase rice yield and resource use efficiency in some contexts [7,18,19]. A combination of site-specific N management and alternate wetting and drying irrigation of super hybrid rice increased grain yield, N partial factor production, and water use efficiency by 12–15%, 27–31%, and 23–27%, respectively, compared with other treatments [14]. In addition, previous studies showed that integrated management practices including N rates, planting densities, water management, tillage depth, and organic fertilizer application improved grain yield and NUE of Japonica rice by optimizing canopy light and N distributions, similar to effects shown in other crops [20,21]. Since the formation of rice yield is affected by many factors, any given management regime may not ensure the yield potential of super hybrid rice. Indeed, many studies have shown that optimization of management to account for genotype by management by environment interactions [12,13,22] increases the likelihood of achieving yield potential and maximum resource use efficiency [23,24]. These studies have shown that the appropriate selection of contextualized genotype by management practices can be conducive to higher yields and resource use efficiency [7,14].

Carbon and N metabolism is one of the most basic metabolic processes related to the synthesis of carbohydrates and proteins in plants [25]. Given the metabolic and energy competition between photosynthetic carbon and N assimilation [26], a valid question may be whether balanced carbon and N metabolism may improve rice yields. The primary N-metabolizing enzymes in plants are nitrate reductase (NR), glutamate dehydrogenase (GDH), glutamine synthetase (GS), and glutamate synthase (GOGAT), whereas the main carbon-metabolizing enzymes in plants are adenosine diphosphate glucose pyrophosphorylase (AGP), sucrose phosphate synthase (SPS), and sucrose synthase [25,26]. Among them, NR is the key enzyme in plant N metabolism, which is the first step reaction catalyzing the conversion of $NO_3^-$ to amino acids and the rate-limiting enzyme for N assimilation in plants [25]. Plants cannot absorb N directly, but they further synthesize the $NO_3^-$ that enters the plant in the form of amino acids through the reduction of NR which is then absorbed and utilized by the plant. Plants catalyze the reaction between glucose-1-phosphate (G1P) and ATP to produce ADPG, which is the direct precursor of starch synthesis and is the main rate-limiting step in plant starch biosynthesis [26]. Increased N fertilization can increase chlorophyll content while maintaining high levels of some enzymes related to

carbon and N metabolism as well as some protective enzyme activities [27]. Furthermore, higher N fertilization can improve the reactive oxygen scavenging ability of leaves, delay leaf senescence, and aid in increasing the late net photosynthetic capacity and promoting photosynthetic product formation along with accumulating substances such as starch and protein [28]. Consequently, appropriate N fertilizer shifting can effectively delay leaf senescence, thereby extending the active time of photosynthesis, enhancing late net photosynthetic capacity, and promoting photosynthetic product accumulation and translocation, which will eventually increase yield [7,10]. Historical improvement of the yield of super hybrid rice has traditionally been more related to the improvement of photosynthetic capacity. It is clear that further research on the impact of photosynthetic physiological characteristics of high-yielding hybrid rice on yield through integrated management is needed to enable improvements in yield and resource use efficiency.

In addition, while the relationship between physiological characteristics and grain yield of rice has been studied extensively, few studies elicit relationships between physiological characteristics and NUE. Given this dearth of information, we aimed to (1) compare differences in the agronomic N use efficiency ($AE_N$), photosynthesis physiology, and carbon and N metabolic properties of super hybrid rice under four representative management systems and (2) elicit relationships between $AE_N$ and the ratio of carbon to N content (C/N) with carbon and N metabolic properties of grain yield in super hybrid rice to better the reasons of high yield and $AE_N$ of super hybrid rice.

## 2. Materials and Methods

### 2.1. Experimental Site and Weather Conditions

From 2017 to 2021, five-year field experiments were conducted at the experimental farm of Yangtze University (112°31′ E, 30°21′ N) in Jingzhou, Hubei Province, China. Soil samples were taken from the upper 20 cm before the initiation of the experiment in each year. In each treatment, we obtained soil samples from the four corners and from the middle and then mixed them together as a single sample to measure soil properties. The average of the four soil samples was then used as the annual soil characteristic value. The soil is calcareous alluvial with pH 6.8, organic matter 18.5 g kg$^{-1}$, alkali-hydrolysable N 110.5 mg kg$^{-1}$, available P 25.0 mg kg$^{-1}$, and available K 105.5 mg kg$^{-1}$ averaged across 2017–2021. In five years, climate parameters including daily minimum and maximum temperatures and solar radiation were collected during the growth period from transplanting to maturity using a Vantage Pro2 weather station (Davis Instruments Corp., Hayward, CA, USA) located near the experimental site (Figure S1). Daily maximum temperatures were 38 °C, 38 °C, 38 °C, 37 °C, and 37 °C, and the daily minimum temperatures were 16 °C, 15 °C, 18 °C, 16 °C, and 17 °C; total solar radiation was 1911 MJ m$^{-2}$, 2124 MJ m$^{-2}$, 2168 MJ m$^{-2}$, 1807 MJ m$^{-2}$, and 2098 MJ m$^{-2}$ during the rice growth season from transplanting to maturity from 2017 to 2021, respectively. The 5-year maximum and minimum temperatures did not differ significantly, and the total solar radiation was higher in 2018, 2019, and 2021 than in 2017 and 2020 during the crop growth season.

### 2.2. Test Material

The rice variety, namely Y-liangyou 900 (YLY900) was used in present study. YLY900 is an *indica* hybrid variety developed by the BioRice (Hunan, Changsha, China) Co., Ltd., with Y58S as the female parent and R900 as the male parent using the two-line method. YLY900 is a representative variety of super hybrid rice with the highest yield of 15 t ha$^{-1}$. In addition, YLY900 is widely planted in southern China and is recommended by the China National Hybrid Rice Research and Development Center in China.

### 2.3. Experimental Design and Crop Management

In order to facilitate management and mechanical transplanting, we adopted a large-area experiment, with each treatment being a large area with an area of about 667 m$^2$. In each year's experiment, we set up four management practices, namely zero-fertilizer

control (CK), farmers' practice (FP), high-yield and high-efficiency management (OPT1) and super-high-yield management (OPT2). Each management practice was an integrated practice based on nitrogen fertilizer, planting density, and water management. In the CK treatment, no fertilizer was used, and the planting density was 30 cm × 18 cm. In the FP treatment, the total N rate was 210 kg N ha$^{-1}$, 70% was applied as basal fertilization and 30% at tillering. Phosphorus at a rate of 90 kg P ha$^{-1}$ as super phosphate was applied as a basal fertilizer, and potassium at 112.5 kg K ha$^{-1}$ as potassium chloride was split equally between the basal and panicle initiation stages in the FP treatment. In addition, the planting density of FP was 30 cm × 18 cm. A total N rate of 195 kg N ha$^{-1}$ was utilized in the OPT1 treatment, and N was applied at the basal, mid-tillering, and panicle initiation stages at a ratio of 5:2:3. The management of phosphorus and potassium fertilizer in the OPT1 treatment was consistent with that in the FP treatment. Additional application of zinc fertilizer in the OPT1 treatment was 5 kg ZnSO$_4$ ha$^{-1}$. In the OPT2 treatment, the total N application rate was 270 kg N ha$^{-1}$. N application at the basal, mid-tillering, panicle initiation, and topdressing phases were 50, 20, 20, and 10%, respectively. The management of phosphorus fertilizer in the OPT2 treatment was the same as that in the FP and OPT1 treatments. Potassium at 150 kg K ha$^{-1}$ as potassium chloride was split equally between the basal and panicle initiation stages in the OPT2 treatment. Furthermore, 5 kg ZnSO$_4$ ha$^{-1}$ and 150 kg Si ha$^{-1}$ was additionally applied in the OPT2 treatment. The planting density of OPT1 and OPT2 was 30 cm × 16 cm. In terms of water management, after transplanting, all treatments were submerged for 5 days and continued with shallow water until 5 days before the panicle initiation stages. Plots were drained for 5 days, flooded at panicle initiation stages, and continuously flooded thereafter until the onset of flowering. The shallow water deep wetting and drying (SWD) [29] method was used for all treatments except FP, which did not receive any irrigation after flowering. For treatments subject to SWD water management, fields were irrigated to a depth of 3.0 cm, allowed to dry, then reirrigated to a depth of 3.0 cm before any visible cracks developed on the soil surface. For OPT1 and OPT2, fields were shallow wetting and drying until one week before harvest, and then dehydrated. Insect and disease infestation was chemically controlled throughout the crop growth cycle. Cultivation management practices were consistent between 2017 and 2021.

Pre-germinated seeds were sown in a seedbed. When 28–32 days old, the seedlings were mechanically transplanted to the field with two seedlings per hill. Transplantation dates were 3 June 2017, 1 June 2018, 10 June 2019, 10 June 2020, and 3 June 2021, which are typical plantings in Jingzhou. Weeds, insects, and diseases were intensively controlled by chemicals to avoid biomass and yield loss.

*2.4. Sampling and Measurements*

The determination of soil properties followed Lu's method [30]. Among them, the pH value was determined using the potentiometric method, soil organic matter content was determined using the potassium dichromate-volumetric method, alkali-hydrolysable nitrogen was determined using the alkali hydrolysis diffusion method, soil available potassium was determined using the flame photometric method, and available phosphorus was determined using the sodium hydrogen carbonate solution-Mo-Sb anti spectrophotometric method.

At the maturity stage, samples were taken at three different places of each treatment as three replicates, and six representative plants from each replicate were sampled to determine the average number of tillers. The sampled plants were divided into straw and panicles, then oven-dried at 105 °C for 30 min, dried at 80 °C until to a constant weight, and then weighed. The number of panicles per hill was calculated to determine the number of panicles per m$^2$. The panicles were hand-threshed, and the filled spikelets were immersed in tap water to separate them from the unfilled spikelets. Three 30-gram samples of filled spikelets and three 3-gram samples of unfilled spikelets were taken to count the number of spikelets. The dry weights of the rachis and spikelets were determined after drying to a

constant weight in an oven at 80 °C. The total dry weight on the ground was calculated as the total dry matter of straw, rachis, and filled and unfilled spikelets. The spikelet and grain filling rate (100 × number of filled spikelets/total spikelets) were also calculated for each panicle. Grain yield was measured in a 5 m$^2$ area per plot and normalized to a moisture content of 0.14 g $H_2O$ g$^{-1}$. All plant fragments were ground into powder using a grinder for the determination of carbon and N content. Total plant tissue N content was determined using a concentrated $H_2SO_4$–$H_2O_2$ disinfection continuous flow analyzer, while carbon content was determined using an elemental analyzer (C0stech ECS4010). N concentrations in stems, leaves, rachis, and spikelets were determined using micro-Kjeldahl digestion, distillation, and titration [31]. The total N content in each plant part was calculated as the product of tissue N concentration and corresponding dry weight. The N content of each plant's parts was summed to obtain the total N content per plant. The N fertilizer efficiency indices were calculated as follows:

The agronomic N use efficiency:

$$(AE_N) = (GY_{+N} - GY_{-N})/FN \text{ (kg grain/kg N)} \tag{1}$$

where FN = N fertilizer applied (kg/ha); GY$_{+N}$ = grain yield of plots that received N fertilizer (kg/ha); GY$_{-N}$ = grain yield of the zero-N control (kg/ha).

Leaf photosynthetic characteristics were measured between 10 and 13 h on 0 and 15 days after the heading stage (15DAH) on flag leaves using the LI6400XT portable photosynthesis measuring system (LI-COR Environmental, Lincoln, NE, USA) connected to a Leaf Chamber Florometer (6400-40, LI-COR, USA) which was used as light source [32]. During the measurement of photosynthesis, leaf temperature was maintained at ambient conditions of temperature (35 °C), PAR (1000 μM m$^{-2}$ s$^{-1}$), and $CO_2$ levels (387 μM mol$^{-1}$). By using this equipment, eight leaf photosynthetic traits were measured viz., photosynthetic rate (P$_N$).

Plants flowering within each treatment were tagged. On 0 and 15 days after the heading stage, the flag leaf blades of tagged plants were sampled, immediately frozen in liquid nitrogen, and stored at −80 °C. These samples were used to analyze the diurnal variations of NR, GS, GOGAT, GDH, AGP, SPS, catalase (CAT), peroxidase (POD), total superoxide dismutase (T-SOD), malondialdehyde (MDA), total ascorbate peroxidase (APX), soluble sugars (SS), and soluble proteins (SP). The measurement methods were as follows:

The NR activity was estimated by using the method described by Hageman and Hucklesby [33]. First, 500 mg of freshly harvested flag leaf tissue was cut into small pieces and transferred into test tubes containing 2.5 mL of 25 mM potassium phosphate buffer (pH 7.5) containing 5 mmol cysteine hydrochloride, 1 mM EDTA, and 1 mM DTT. The homogenate was centrifuged for 45 min at 10,000 rpm at 4 °C. A volume of 0.3 mL of supernatant was used immediately for enzyme assay. An enzyme extract and reaction mixture containing 0.1 mol potassium phosphate buffer (pH 7.5), 0.1 mol potassium nitrate, and 14 mg/10 mL of NADH were incubated at 33 °C for 30 min. After incubation, 1 mL of sulphanilamide and 1 mL of NEDD reaction were stopped with the addition of zinc acetate. The nitrite produced was estimated at 540 nm using the Spectrascan UV2600 (Toshniwal Instruments (Chennai, India) Pvt. Ltd.). The enzyme activity was expressed as μM/ (g$^{-1}$ h$^{-1}$).

The extraction and activity determination of GS, GOGAT, GDH, AGP, SPS, CAT, POD, T-SOD, MDA, APX, SS, and SP activity or content followed Zou [34]. Crude enzyme fluids could be used to determine the activities of GS, GOGAT, and GDH. The measurement of GS activity was conducted following Rhodes [35]. The determination method of GOGAT activity was conducted following Singh and Srivastava [36], while GDH activity was measured following Loulkakais and Ronbelakis-Angelakis [37]. AGP activity was measured in the direction of ADP-Glusynthesis at 37 °C, according to Ghosh and Preiss [38]; SPS activity was measured using methods described by Okamura [39]. CAT activity was determined by following the consumption of $H_2O_2$ (extinction coefficient 39.4 mM$^{-1}$ cm$^{-1}$) at 240 nm for 1 min [40]. POD activity analysis was carried out as described by Zou [34]. T-SOD activity was assayed by monitoring the inhibition of photochemical reduction of

nitroblue tetrazolium (NBT) according to the method of Giannopolitis and Ries [41]. Lipid peroxidation was detected by measuring the MDA content in 0.5 g FW of flag leaves following Heath and Packer [42] with a slight modification. APX activity was measured using the methods described by Nakano and Asada [43]. The soluble protein content was measured using the methods described by Bradford [44]. Soluble sugars were quantified according to the phenol–sulfuric acid method of Pistocch [45] using glucose as a standard.

### 2.5. Data Analysis

Data were analyzed using analysis of variance (Statistix 8, Analytical Software, Tallahassee, FL, USA); genotypic means were compared using least significant differences (LSD) with a significance level of 0.05 unless stated otherwise.

## 3. Results

### 3.1. Grain Yields

Grain yields were significantly different among the four treatments; ranked OPT2 > OPT1 > FP > CK (Figure 1). Average grain yields of OPT2, OPT1, FP, and CK from 2017 to 2021 were 11.2, 9.7, 8.6, and 6.7 t ha$^{-1}$, respectively. Over five years, average grain yield of OPT2 was 15% and 30% higher than OPT1 and FP, respectively.

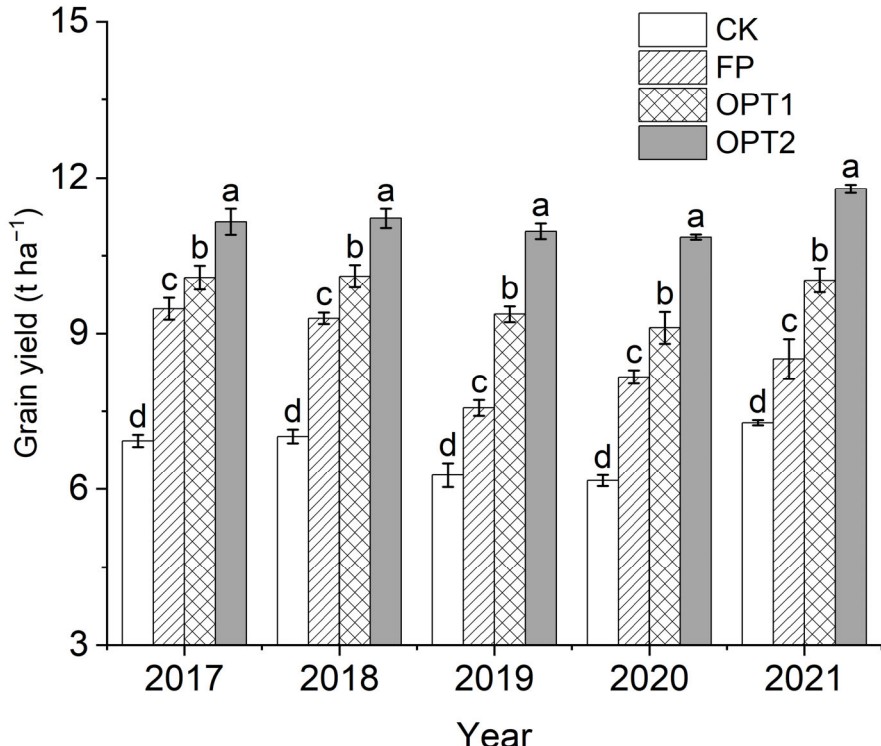

**Figure 1.** Grain yield of four management systems from 2017 to 2021 in Jingzhou, Hubei Province, China. Means followed by the same letter do not significantly differ (LSD, *p* < 0.05).

### 3.2. N Use Efficiency

The AE$_N$ varied considerably among treatments, and the trend was generally consistent across years, with OPT2 > OPT1 > FP, from 2017 to 2021 (Figure 2). The average AE$_N$ of OPT2 was 17 kg kg$^{-1}$, which was 10 and 78% higher than OPT1 and FP, respectively.

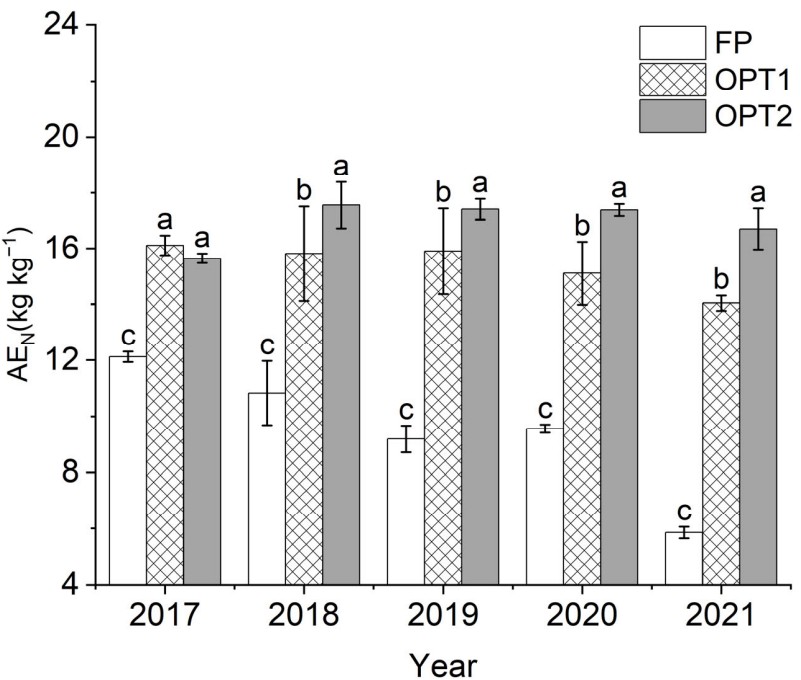

**Figure 2.** Nitrogen agronomic efficiency of four management systems from 2017 to 2021 in Jingzhou, Hubei Province, China. Means followed by the same letter do not significantly differ (LSD, $p < 0.05$).

### 3.3. The Photosynthetic Rate in Flag Leaves

A significant disparity was observed in the photosynthetic rates at 15 days before the heading stage (15DBH), at HD, and at 15DAH in different cultivation patterns ($p < 0.05$), and all showed OPT2 > OPT1 > FP > CK (Figure 3). Among them, OPT2 at HD had the highest mean $P_N$ of 53.5 $\mu$mol m$^{-2}$ s$^{-1}$, which was 92%, 53%, and 19% higher than that of CK, FP, and OPT1, respectively.

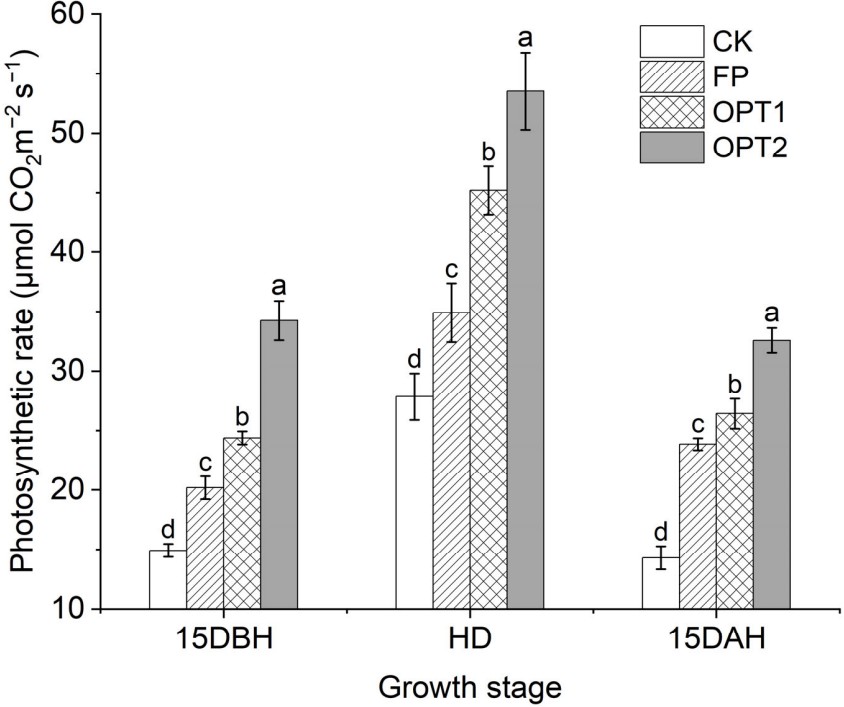

**Figure 3.** The photosynthetic rate of flag leaves under four management systems at 15 days before the heading stage (15DBH), heading stage (HD), and 15 days after the heading stage (15DAH). Means followed by the same letter do not significantly differ (LSD, $p < 0.05$).

### 3.4. The NR, GS, GOGAT, and GDH Enzyme Activities

At HD, the NR, GS, and GOGAT enzyme activities of OPT2 were higher than in the other treatments (Figure 4). Among them, the NR enzyme activity of OPT2 had increased by 29% and 87% compared to OPT1 and FP, and the NR enzyme activity of OPT1 was 44% higher than FP, respectively. The GS and GOGAT enzyme activities of OPT2 were the highest across treatments, which were 18% and 29% higher than OPT1, and 42% and 197% higher than FP, respectively. In addition, the GDH enzyme activity of OPT1 increased by 65% and 18% for FP and OPT2, respectively.

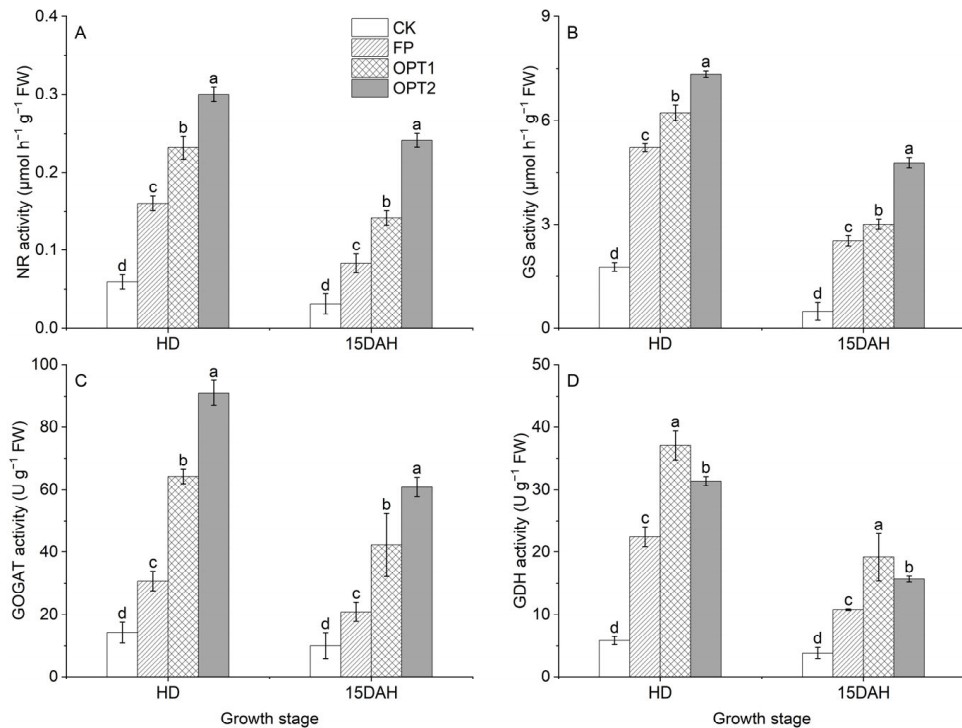

**Figure 4.** The nitrogen metabolism-related enzyme activity of NR (**A**), GS (**B**), GOGAT (**C**) and GDH (**D**) under four management systems at HD and 15DAH. Means followed by the same letter do not significantly differ (LSD, $p < 0.05$).

The activities of four N-metabolizing enzymes decreased significantly at 15DAH in the four treatments compared with those at HD. The NR enzyme activities at 15DAH were decreased by 48% (CK), 48% (FP), 39% (OPT1), and 20% (OPT2) compared to that at HD, respectively. The GS and GOGAT enzyme activities at 15DAH were significantly decreased in OPT2 by 35% and 33%, OPT1 by 52% and 34%, FP by 73% and 30%, and CK by 52% and 32% compared to that at HD, respectively. In addition, the GDH enzyme activity at 15DAH decreased significantly compared to that at HD, with 34%, 52%, 48%, and 50% decreased in the CK, FP, OPT1, and OPT2 treatments, respectively.

### 3.5. The Activities of AGP and SPS Enzyme

The AGP and SPS enzyme activity differed significantly among the four treatments (Figure 5). At HD, the AGP activity of OPT2 was 43% and 47% higher than OPT1 and FP, respectively. Similar to AGP, the SPS activity of OPT2 had increased by 30% and 95% compared to OPT1 and FP, respectively.

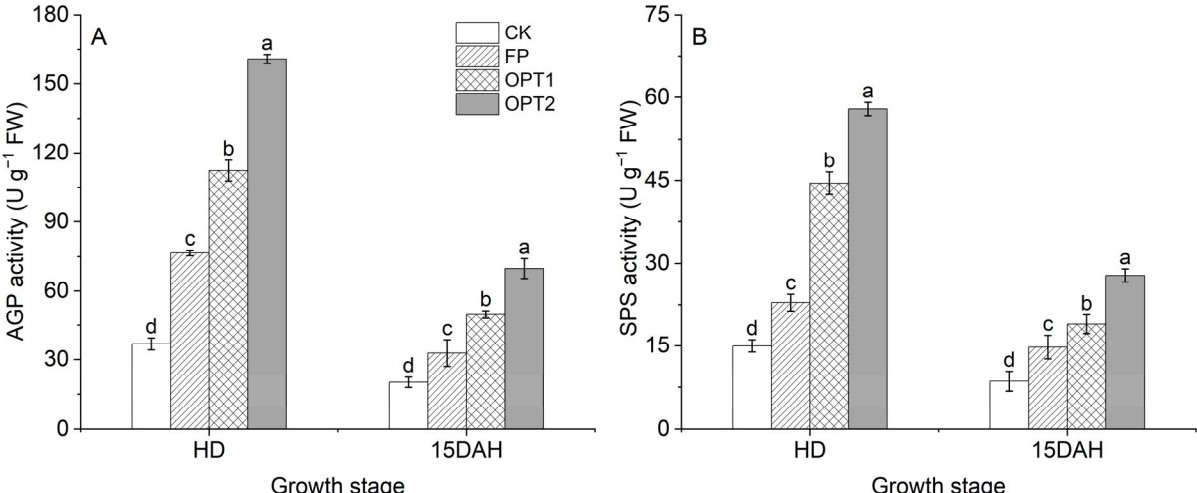

**Figure 5.** The carbon metabolism-related enzyme activity of AGP (**A**) and SPS (**B**) under four management systems at HD and 15DAH. Means followed by the same letter do not significantly differ (LSD, $p < 0.05$).

At 15DAH, the AGP enzyme activities had significantly decreased by 56.6% and 52.0% (under OPT2), 56% and 58% (OPT1), 57% and 36% (FP), 45% and 43% (CK), respectively.

*3.6. C/N in Straw and Grains*

The C/N in straw and grains differed significantly across treatments and trends were mostly consistent across years (Figure 6). The average C/N in the straw of OPT2 had decreased by 23% and 35.5% compared to OPT1 and FP across 2017–2021, respectively. The C/N in grains of OPT2 had decreased by 15% and 36% compared to OPT1 and FP when averaged across five years, respectively.

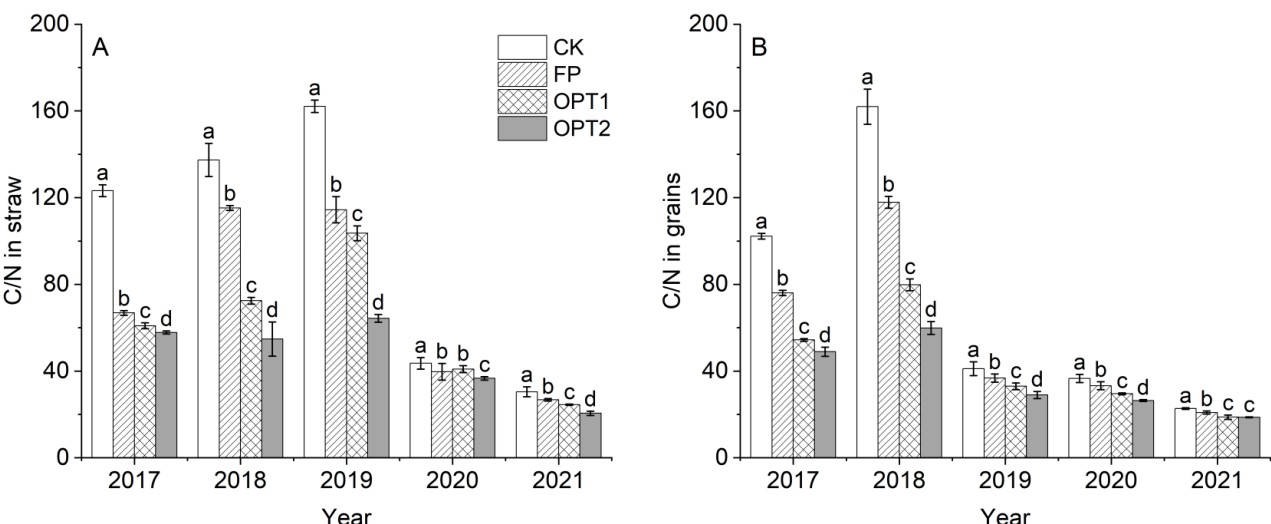

**Figure 6.** C/N in straw (C/N$_S$) (**A**) and C/N in grains (C/N$_G$) (**B**) of the plants under four management systems. Means followed by the same letter do not significantly differ (LSD, $p < 0.05$).

*3.7. Soluble Sugars and Soluble Proteins*

The SS and SP of OPT2 were significantly higher than OPT1, FP, and CK (Figure 7), and they were not significantly different between the FP and OPT1 treatments. The SS at 15DAH increased significantly across treatments compared to that at HD, by 71%, 77%, 82%, and 75% in CK, FP, OPT1, and OPT2, while the SP content decreased significantly by 43%, 42%, 42%, and 42%, respectively.

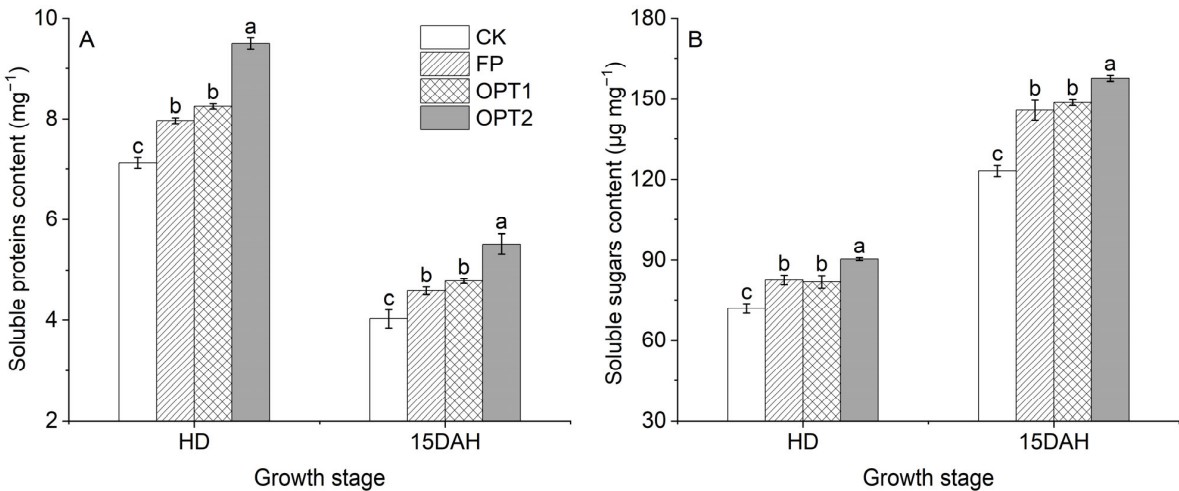

**Figure 7.** Soluble sugars (**A**) and soluble proteins (**B**) under four management systems at HD and 15DAH. Means followed by the same letter do not significantly differ (LSD, $p < 0.05$).

*3.8. The Activities of POD, CAT, T-SOD, and APX Enzymes and the Content of MDA*

At HD, the POD and CAT enzyme activities showed a consistent trend across treatments, with OPT2 > OPT1 > FP > CK (Figure 8), but the T-SOD enzyme activities were not significantly different between FP and OPT1. Among them, the POD and CAT enzyme activities of OPT2 were 17% and 21% higher than OPT1, and 38% and 49% higher than FP, respectively.

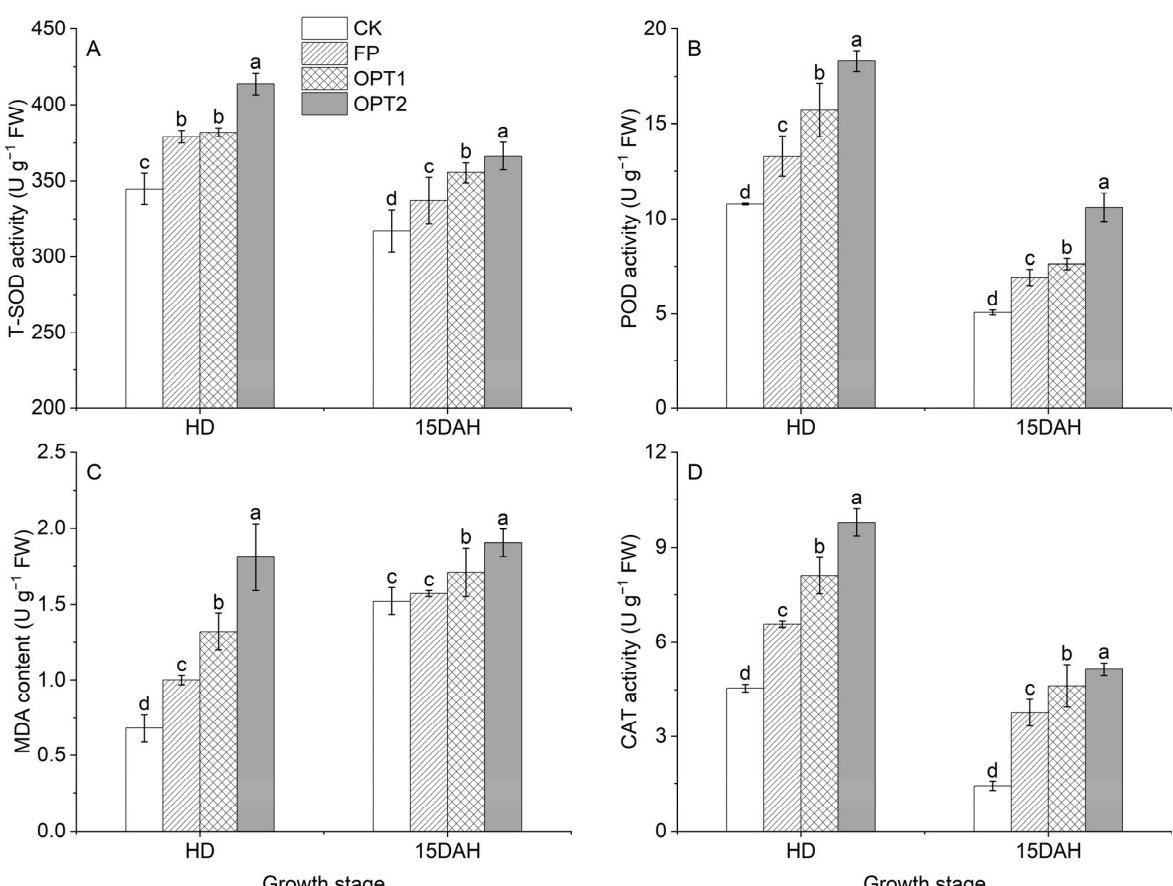

**Figure 8.** The Leaf protective enzyme system including T-SOD (**A**), POD (**B**), MDA (**C**) and CAT (**D**) under four management systems at HD and 15DAH. Means followed by the same letter do not significantly differ (LSD, $p < 0.05$).

At 15DAH, the activities of T-SOD, POD, and CAT were significantly decreased compared to HD. The T-SOD enzyme activities of OPT2 had decreased by 11%, which was higher than that of OPT1 and FP. Moreover, the POD activity had decreased by 42% and 52% in OPT2 and OPT1, respectively.

In addition, compared to HD, the MDA content in all treatments increased significantly at 15DAH. The MDA content at 15DAH was increased by 124% (CK), 57% (FP), 30% (OPT1) and 18% (OPT2) compared to HD, respectively. The APX enzyme activity of OPT2 was the highest across the four treatments. Moreover, the APX enzyme activity significantly increased at 15DAH, and the APX enzyme activity differed significantly across treatments (Figure 9). The APX enzyme activity at 15DAH of different treatments increased by 280% (CK), 492% (FP), 236% (OPT1) and 122% (OPT2) compared to HD, respectively.

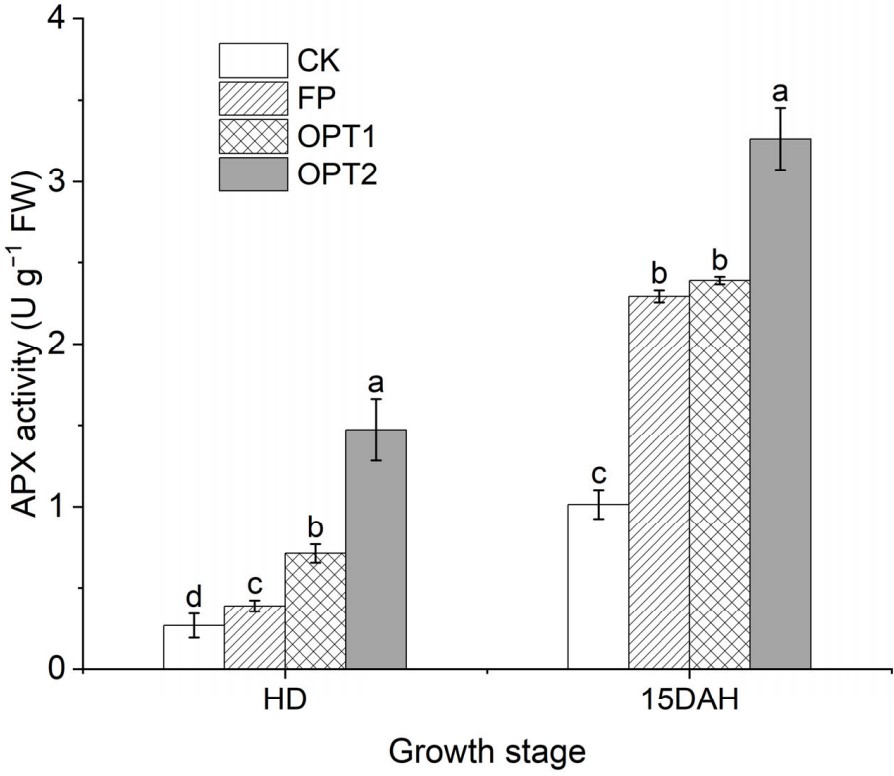

**Figure 9.** The APX activity under four management systems at HD and 15DAH. Means followed by the same letter do not significantly differ (LSD, $p < 0.05$).

### 3.9. Relationships between N Use Efficiency and the Physiological Characteristics of Flag Leaves at HD and 15DAH

A significant correlation was observed between AGP, SPS, NR, GS, GDH, and GOGAT with $AE_N$ (Figure 10). At HD, $AE_N$ was highly correlated with AGP, NR, and GS, with R values of 0.93, 0.95, and 0.93, respectively. In addition, $AE_N$ was positively correlated with CAT and POD, while was negatively correlated with T-SOD, with R values of 0.91, 0.92, and −0.91, respectively. Moreover, $AE_N$ was also negatively correlated with soluble sugars and $C/N_G$, with R values of −0.97 and −0.92, respectively. At 15DAH, $AE_N$ was highly correlated with AGP, SPS, NR, and GS, with R values of 0.92, 0.92, 0.96, and 0.98, respectively.

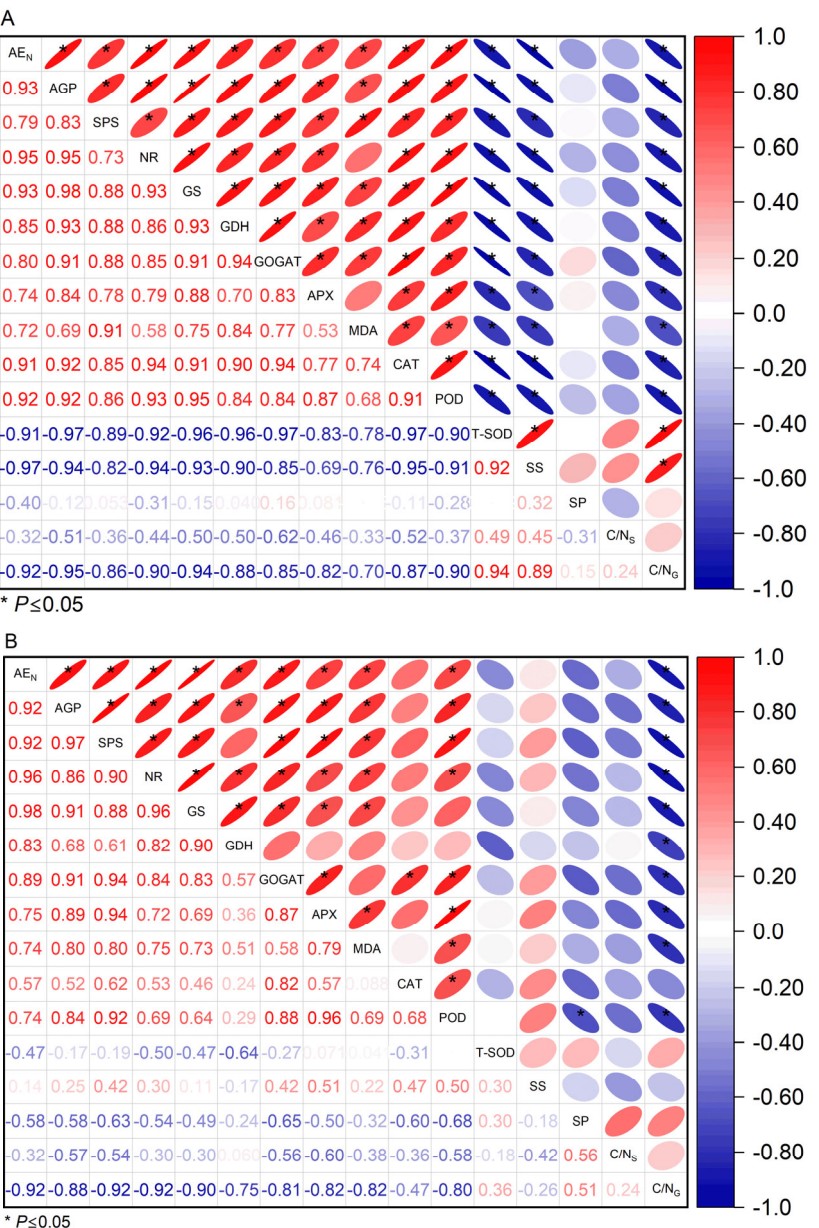

**Figure 10.** Relationship between the agronomic nitrogen use efficiency (AE$_N$) and adenosine diphosphate glucose pyrophosphorylase (AGP), sucrose phosphate synthase (SPS), nitrate reductase (NR), glutamine synthetase (GS), glutamate dehydrogenase (GDH), glutamate synthase (GOGAT), total ascorbate peroxidase (APX), MDA, catalase (CAT), peroxidase (POD), total superoxide dismutase (T-SOD), Soluble sugars (SS), soluble proteins (SP), C/N$_S$ and C/N$_G$ at HD (**A**) and 15DAH (**B**). * denote statistical significance at the 0.05 levels.

## 4. Discussion

### 4.1. Effect of Optimal Management on Grain Yield and N Use Efficiency of Super Hybrid Rice

To improve rice growth, yield, and resource use efficiency, various management practices have been historically proposed [46]. The plant growth and grain yield in rice can be regulated through N fertilizer, transplanting density, and water management [47]. Numerous studies have shown that increasing planting density with reduced N input can enhance rice yield and NUE [7,48]. In this study, the grain yield of OPT2 averaged across five years was 11.2 t ha$^{-1}$, which was 15% and 30% higher than that of OPT1 and FP, respectively. OPT2 improved N translocation patterns, which postponed N application to late-season phenology, delayed leaf senescence, extended the filling period, and promoted

the formation of high sink potential [7]. In addition, OPT2 also increased the total panicles number per unit area by increasing the transplanting density, therefore, improving the yield. Compared with OPT2 and OPT1, the reasons for the lower yield in FP might be, on the one hand, that the effective panicle number per unit area is not sufficient owing to insufficient transplanting density. On the other hand, farmers tend to pay attention to the application of basal fertilizer and tiller fertilizer but overlook the application of PI and topdressing in the late rice growth stages. As a result, super rice varieties could not take advantage of large panicles to improve grain yield in FP treatment which suggests that under FP, N management might be suboptimal or poorly timed. This suggests an opportunity for us or for future research to produce 'best management practice' guidelines for farmers by region.

Alternating wet and dry irrigation can reduce the occurrence of ineffective tillers in the initial stage of rice, promote spikelets formation and grain filling, and ultimately increase rice yield and water use efficiency [29]. Better management techniques, consisting of alternate wet and dry irrigation and increased density, have been reported to considerably increase rice yields and decrease N fertilizer use [32,48]. In our study, compared with FP, alternating wet and dry irrigation patterns were carried out during the critical period of OPT2 and OPT1 rice growth, which saved water resources and significantly increased rice yield. Some studies have confirmed that increasing N application can reduce yield loss due to high temperature during flowering in super hybrid rice [49,50]. Therefore, rice yield can be greatly improved with reasonable adjustment of N fertilizer transport pattern, moderate dense planting, and optimized irrigation pattern.

In our study, the $AE_N$ of FP was 10.1 kg kg$^{-1}$, and the average annual $AE_N$ of OPT2 and OPT1 had increased by 63% and 51% compared to FP, respectively. The analysis indicated that the application of N fertilizer under FP was categorized into only two applications, i.e., the base fertilizer applied prior to transplanting and the tiller fertilizer applied 7 days after transplanting. The effect of tiller fertilizer on rice lasted until about 14 days after fertilizer application, and the effect of fertilizer during PI lasted merely 10 days [51]. It is obvious that the fertilizer applied in FP can only work during the early stage of rice growth and development. However, the fastest rate of N uptake by rice is at PI which determines the critical period of spikelet differentiation and degeneration. Thus, FP had low NUE because plants only obtain the required N from the surrounding environment in the soil during the middle and late stages of rice growth. Compared with FP, the OPT2 and OPT1 in the current study reduced the amount of basal and tillering fertilizers applied, and fertilized at PI and topdressing phases, and thus improved rice NUE. In addition, previous studies have shown that N fertilizer management in combination with other cultivation techniques can significantly improve the $AE_N$ and rice yield [7,52]. For example, the rational addition of N fertilizer with organic fertilizer not only reduces the amount of chemical fertilizer applied but also enhances the supply capacity of the soil N for rice growth [50,53]. Therefore, an optimized cultivation pattern can realize the synergistic improvement of grain yield and NUE in super rice production.

### 4.2. Effect of Optimal Management on Physiological Characteristics of Super Hybrid Rice

Our results showed that carbon- and N-metabolizing enzyme activity in super hybrid rice leaves were highest in OPT2, followed by OPT1. The high-yielding super rice population was capable of metabolizing large amounts of carbon and N, which facilitated the accumulation and translocation of dry matter and improved rice yield [54]. In contrast to FP, OPT2 and OPT1 patterns greatly boosted the activities of N-metabolizing enzymes such as NR, GS, GOGAT, and GDH and carbon-metabolizing enzymes such as AGP and SPS in leaves at HD and 15DAH. The leaf N metabolism was accelerated by the increased activities of N-metabolizing enzymes [33]. Moreover, the enhanced N metabolism could promote carbon metabolism, which would then enhance the activities of AGP and SPS enzymes [31,55]. Postponing N application facilitates post-flowering N accumulation and increases the photosynthetic rate of flag leaves, while also increasing the activities of key

enzymes of carbon and N metabolism [19], such as NR, GS, RuBP case, and SPS, for the purpose of promoting post-flowering N accumulation and translocation, which is the primary regulatory pathway to achieve simultaneous improvement of yield and NUE. Most of the soluble proteins in plants are enzymes involved in diverse metabolisms, while soluble sugars can act as structural substances of cells and supply energy for N metabolism by accelerating breakdown under the action of carbon metabolism enzymes [31].

We found that as rice growth progresses after heading, activities of protective enzyme systems such as T-SOD, POD, and CAT in leaves declined and MDA content increased. This intensified leaf senescence and caused the photosynthetic rate of rice to drop. The study also revealed that the optimized cultivation pattern substantially increased the net photosynthetic rate values of flag leaves at HD, and the decrease with the reproductive period was also significantly lower compared with FP. The reason could be the increased N application of PI and topdressing phases at the later stages of OPT1 and OPT2, which caused a slower decrease in their photosynthetic rates. Studies have shown that the leaf photosynthetic activity in the late-growth period is more likely to sustain a constant rate with increased N nutrition [56]. Therefore, proper postponement of N application is beneficial to maintain the activity of protective enzymes, delay leaf senescence, and ensure photosynthesis of leaves during the grain filling stage, and further increase grain yield.

*4.3. Optimization of Management to Increase Yield and N Use Efficiency through Enhanced Carbon–N Metabolism*

Grain N accumulation is a key indicator of N uptake and utilization, and these two components are more important to improve plant N accumulation [57]. Compared to FP, C/N in straw and C/N in grains showed a tendency to decrease under OPT1 and OPT2. Furthermore, this trend is consistent with the NUE under different cultivation patterns. This indicates that the optimized cultivation patterns can significantly increase the $AE_N$ and N content in plants. However, plants cannot absorb N directly and must convert it into $NH_4^+$ which can be absorbed by plants through N metabolic pathways [58]. In addition, N metabolism could promote carbon metabolism to meet the requirements of N metabolism for energy and carbon skeleton. Adequate N nutrition is a prerequisite for accelerating photosynthesis, increasing carbon sources, and promoting carbon metabolism.

In this study, correlation analysis indicated that $AE_N$ under the four cultivation patterns was highly correlated with the carbon and N metabolism-related enzymes along with related substances at HD and 15DAH. Among them, AGP and SPS, NR and GS as well as CAT and POD had high correlations. Stronger carbon and N metabolism is conducive to the transfer and accumulation of rice materials, which is an important guarantee of a high yield of rice [59,60]. As rice growth progresses after heading, the activity of protective enzymes such as T-SOD, POD, and CAT in rice leaves decreases, the leaves become more senescent, and the activity of enzymes related to carbon and N metabolism decreases. And as the leaves senesce, the activity of the protective enzyme system consisting of SOD–POD–CAT in rice leaves decreases, leading to an increase in MDA content. Appropriate N fertilization setback can maintain a certain activity of protective enzymes during the grain-filling stage, inhibit the increase of MDA content, and delay the senescence of leaves. In addition, the optimized cultivation pattern can delay the senescence of the root system, ensure a certain capacity of material uptake and transfer in the late reproductive stage of rice, and increase yield and NUE [61]. It also ensures a certain photosynthetic capacity of functional leaves in the late stage of rice, which is conducive to the accumulation and translocation of rice materials, thus obtaining high yield and high NUE.

This study indicates that optimizing cultivation patterns can significantly improve the carbon and N metabolism capacity of plants as well as the anti-aging mechanism, which is important for improving rice yield and NUE. In our experiment, OPT1 and OPT2 that incorporated field N fertilizer management showed a higher magnitude of improvement in yield and $AE_N$ compared to FP. Therefore, a synergistic improvement in rice yield and $AE_N$ was achieved by enhancing leaf carbon and N metabolism of super hybrid rice.

## 5. Conclusions

In conclusion, compared with the farmers' practice, optimized integrated management practices including N fertilizer and water management, appropriate dense planting significantly increased grain yield and NUE though enhanced post-flowering source production capacity resulting from higher leaf carbon- and N-metabolizing enzyme activity and delaying leaf senescence.

**Supplementary Materials:** The following supporting information can be downloaded at: https://www.mdpi.com/article/10.3390/agronomy13010013/s1, Figure S1: Daily maximum temperature, minimum temperature, and solar radiation from transplanting to maturity in 2017 (A), 2018 (B), 2019 (C), 2020 (D) and 2021 (E) in Jingzhou, Hubei Province, China.

**Author Contributions:** Y.Z. designed the experiments and revised the paper; J.D., X.Z., C.W., and J.Y. investigated the traits, J.D. analysed the data, J.D. wrote the manuscript, K.L., M.T.H., L.H., and X.T. aided with conceptualization, scientific rigor, and manuscript editing. All authors have read and agreed to the published version of the manuscript.

**Funding:** This study was funded by the National Natural Science Foundation of China (Grant No. 32001467 and 32172108).

**Data Availability Statement:** The original contributions presented in the study are included in the article/Supplementary Material, further inquiries can be directed to the corresponding author/s.

**Acknowledgments:** We are grateful to Yangtze University Excellent Doctoral Dissertation Development Program.

**Conflicts of Interest:** The authors declare that the research was conducted in the absence of any commercial or financial relationships that could be construed as a potential conflict of interest.

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
