# Peer review of "Optimizing Agronomy Improves Super Hybrid Rice Yield and Nitrogen Use Efficiency through Enhanced Post-Heading Carbon and Nitrogen Metabolism"

_agronomy, doi:10.3390/agronomy13010013_

Round 1

Reviewer 1 Report

Dear Editors staff,

The authors have conducted a four treatments experiment (management practices) (i.e., zero-N control, CK; farmer practice, FP; high yield and high efficiency management, OPT1 and super high yield management, OPT2) over five years in order to enhance post-heading carbon and nitrogen metabolism in super hybrid rice.

The topic is very interesting and the manuscript is very well written. All parts of the research are well written and the introduction, results and discussion are at a high level except for the part related to materials and methods, especially experimental design and irrigation.

Despite the importance of the topic at the present time and the work and efforts done by researchers in both field and labs for five years, I have some comments:

·         I am wondering what is the type of the experimental design you used, is it RCBD, combined ANOVA in RCBD?

·         What is the amount of water you used in each treatment?

·         Did you calculate the water use efficiency?

·         Different amount of water may affect the final yield of each treatment?

·         Is this cultivar ''Y- liangyou900'' the only one recommended for cultivation of the investigated area?

·         Do you expect that is there a genetic variation among cultivars if you applied the same treatments?

·         Do you expect that you can find a different cultivar ''Genotype'' may response positively than Y- liangyou900?

·         The recommendations of current study are quite good, but in my point of view as breeder, I have to test another rice genotypes.

Generally, the manuscript can be accepted

Reviewer 2 Report

Dear authors,

Thank you for the opportunity to review this Manuscript (Increasing yield grains through management practices that enhance post-heading carbon and nitrogen metabolism in super hybrid rice). The study has great results and demonstrates that the differences in the agronomic N use efficiency (AEN), photosynthesis physiology, and carbon and N metabolic properties of super hybrid rice under four representative management systems and (2) elicit relationships between AEN and the ratio of carbon to N content (C/N) with carbon and N metabolic properties of grain yield in super hybrid rice to better the reasons of high yield and AEN of super hybrid rice. There is some aspect that should be reviewed by authors, but the Manuscript is well-written.

ABSTRACCT

What are APT1 and APT2? AGP and SPS? NR, GS and GOGAT? And others in the abstract

INTRODUCTION

Line 49, give examples of N loss and environmental pollution

Line 76, explain NUE first

MATERIAL AND METHODS

For the treatments is difficult to identify which management positively influence the yield.

The authors could explore economic viability of treatments with high fertilizer rates

Explain the influence of climate on yield in each year.

There is no “conclusion”. The authors repeat the objective

Give more details about this genetic material.

Reviewer 3 Report

The presented work: “Increasing yield grains through management practices that enhance post-heading carbon and nitrogen metabolism in super hybrid rice” addresses the very important issue of how rice yields vary as a result of fertilisation, and presents the relationships affecting yields. The authors present the important issues of photosynthesis and the activity of N- and C-metabolising enzymes. I believe that the research material presented in the paper is fully sufficient, the conclusions are correctly formulated and the literature cited is appropriate.

However, I believe that several aspects of the work need refinement:

Line 50: The authors emphasise the impact of excessive N use on environmental pollution. This is a very important piece of information, but I think it would also be useful to add information on the over-application of N on the economic profitability of rice cultivation.

The Materials and methods section needs to be completed:

2.1 Experimental Site and Weather Conditions

Add information on how soil samples were taken and the total number of samples taken in each year. Was one sample taken from the entire study area or one for each plot? Also information on which methods were used for soil analyses should be added.

Description of weather conditions very sketchy, I suggest describing in more detail with consideration of key periods for rice cultivation.

2.3 Experimental design and crop management

I believe that additional information should also be provided in this section. How was the planting carried out? Was the experiment carried out in replicates in the year in question? Given the given standard deviation in yield probably yes, this would need to be included in the description. What was the total number of experimental plots? What was the area of each plot?

There is also no information on how much potassium and phosphorus fertiliser was applied.

Lines 175 - 177 should be moved to section 2.1 Experimental Site and Weather Conditions.

3 Results

In the description of the research, a number of places mention average values and significant differences between them, but this information is missing from the figure. For example, in grain yield, the results from individual years are shown on the figure, while the description gives average values for individual treatments. I suggest adding on the individual bars of the figure their values together with the standard deviation, this would greatly improve the readability of the figure. Additionally, I believe that the figure would become more readable if colours rather than shades of grey were used. (This is only a suggestion). The percentage increases for each treatment are given as an approximation so I suggest adding: "ca."

4 Discussion

Line 394 - 394. The authors mention the effect of N fertilisation in mitigating yield losses due to high temperatures during flowering. Is this also supported by the authors' findings, not just the literature?

Additionally, I noticed variable citations in the text at several places in the paper. Three types occur: Author et al. [year] [number from References section]; Author et al. [year]; Author et al. [number from References section] this should be standardised as required by the publisher.

Round 2

Reviewer 3 Report

Dear Authors,

Thank you for the changes you made and for responding to the review questions.

Introduction

"However, as the production technology of nitrogen fertilizer is relatively mature, the price of nitrogen fertilizer can be accepted by farmers, so farmers prefer to apply more fertilizer to achieve high yield."

In many countries around the world, current mineral fertiliser prices account for a significant proportion of production, hence my suggestion. I agree with the statement that when the price of mineral fertiliser is relatively low, farmers apply very large amounts of fertiliser. Unfortunately, this very often leads to environmental costs, some of which the authors have mentioned. An additional significant impact of using large amounts of mineral fertiliser is also the ingress of heavy metals into the soil which the fertiliser may contain.

Line 53 uses a different form of citation: Peng et al. (2009); should be corrected

Thank you for completing my suggestion. Considering the additional changes made in the Introduction section, I believe it is well written and fulfils its role in the content of the manuscript.

Materials and methods

Thank you for the clarification and changes made regarding plot area, soil sampling rates and phosphorus and potassium fertilisation.

I find the Authors' response to the more detailed description of weather conditions appropriate and reasonable.

However, I do not fully agree with the statement: "Given the annual meteorological data of the growing season can reflect the general climate conditions of the whole growing period of rice (...)" (response to the review)

Because high temperature during a critical demand period (e.g. usually the flowering period) can result in increased infertile flowers and poorly formed seeds, and inhibit plant growth, shorten the maturation period and ultimately significantly reduce yield. 

Of course, this does not apply to the work presented here; according to the authors' response, the research mainly focuses on yield differences from a physiological perspective. Additional considerations of the influence of weather conditions at particular periods of plant development would have resulted in a considerable expansion of the manuscript and loss of focus on the most important aspect of the work, i.e. carbon and nitrogen metabolism, given by the Authors.

The Materials and Methods section has not been completed with the methods used to analyse the soil samples collected. Completion of this aspect will allow the methodology used to be fully presented and will allow the experiment to be repeated in another area or the application of the methods in practice. Please complete the method used to analyse the soil samples.

Line 303, 305, 306, 317 and 3019 use a different form of citation, should be corrected.

Results

Thank you for responding to the review, I consider the response to be legitimate. In addition, the changes made improve the quality of this section. I believe the Results section is adequately presented.

Discussion

"Unfortunately, we did not do trials related to affecting rice yield about the effect of N fertilizer in mitigating yield losses due to high temperatures during flowering in this experiment"

Thank you for responding to the suggestion. In the submitted manuscript, there is scope to analyse the mitigation of weather conditions through N fertilisation, but as mentioned earlier this will result in a significantly expanded manuscript. It is reasonable to focus on carbon and nitrogen metabolism as this is the main aspect of the work.

Conclusions

The changes made provide a more synthetic summary of the manuscript. I consider the change made to be very appropriate.

Assessing the whole manuscript, I find it very interesting and adequately written. With the addition of soil sample analysis methods and minor editorial corrections, it can be accepted for publication.
